

# Gender differences on medical students' attitudes toward patient-centred care: a cross-sectional survey conducted in Heilongjiang, China

Wei Liu[1,*], Yanhua Hao[1], Xiaowen Zhao[2], Tao Peng[3], Weijian Song[1,4,*], Yuxin Xue[1,5], Siyi Tao[1], Zheng Kang[1], Ning Ning[1], Lijun Gao[1], Yu Cui[1], Libo Liang[1] and Qunhong Wu[1]

[1] Department of Social Medicine, School of Health Management, Harbin Medical University, Harbin, China
[2] Department of Health Economics, School of Health Management, Harbin Medical University, Harbin, China
[3] Department of Sexual Health Education, School of Health Management, Harbin Medical University, Harbin, China
[4] Department of Humanities and Social Sciences, Harbin Medical University, Daqing, China
[5] Chengyang People's Hospital, Qingdao, China
* These authors contributed equally to this work.

## ABSTRACT

**Objectives**. Assessing medical students' attitudes toward patient-centred care is essential to bettering medical education. Based on doctor-patient relationships and the medical system in China, it is important to explore the impact of gender differences and other background factors on patient-centred attitudes and to provide references for medical education reform.

**Methods**. A cross-sectional study was conducted on fourth-year medical undergraduate students from November 2017 to March 2018 in Heilongjiang Province, Northeast China. The Chinese-revised Patient-Practitioner Orientation Scale (CR-PPOS), which has been validated in previous research, was used to measure the medical students' attitudes. The medical students' demographic data was collected, including their gender, age, information on whether they have siblings, family residence location, doctor(s) for parents, year in which the student first experienced clinical practice, and student category.

**Results**. A total of 513 students (91.12%) completed the survey. The Chinese medical students scored considerably higher for 'Caring' (including patients' preferences into the decision-making process) than for 'Sharing' (sharing information/responsibility with patients). These students tended to have patient-centred attitudes, as measured by an average overall CR-PPOS score of 3.63 (scores higher than 3.5 indicate patient-centred attitudes), which is higher than Malian (3.38) and Pakistani (3.40) medical students but lower than American (4.57) and Brazilian (4.66) students. Female students ($P < 0.05$) were significantly associated with more patient-centred attitudes and with higher 'Sharing' and 'Caring' subscale scores. *Student category* ($P < 0.05$) was associated with 'Sharing' and 'Caring' scores. *Clinical hospital students* ($P < 0.05$) were associated with more patient-centred attitudes and with higher 'Sharing' and 'Caring' subscale scores, *Students without siblings* ($p < 0.07$) were associated with the higher 'Sharing' subscale scores.

Corresponding authors
Libo Liang, llbhit@163.com
Qunhong Wu, wuqunhong@163.com

**Conclusions**. In China, gender has a significant impact on medical students' patient-centred attitudes, which is similar to findings from other countries. If medical schools want to raise patient-centred attitudes across the board and bridge the gap between male and female patient-centred attitudes, gender, student category, and other factors should be incorporated into medical education.

# INTRODUCTION

Traditional biomedicine practices are based on Western science, focusing on the specific disease rather than the patient as a whole and tending to grant doctors the decision-making power (*Engel, 1980*). Patient-centred care, which was proposed by Balint et al. and is an overall approach to medical practice compared with the biomedical medical practice, treats patients as unique human beings and establishes a more egalitarian relationship between doctor and patient (*Balint, 1969*; *Ishikawa, Hashimoto & Kiuchi, 2013*). Byrne et al. and Levenstein et al. further expanded upon these ideas, which have attracted the attention of medical practices over the past few decades (*Levenstein et al., 1986*). Patient-centred care has been recognized as an indispensable factor that is vital to improving the quality of healthcare delivery, patient care strategies, and medical education (*Aljuaid et al., 2016*).

Patient-centred care refers to the establishment of relationships among doctors, patients, and their families in order to care for patients' needs and preferences and to provide the necessary information and support so that patients can actively participate in clinical decision-making and in their own care (*America, 2001*; *Health & Delivery, 2001*). With this approach, doctors use their communication skills to understand the patient's ideas, expectations, emotions, preferences, and concerns about the illness in order to seek an integrated understanding of the patient's world (*Stewart, 2001*; *Epstein et al., 2005a*; *Epstein et al., 2005b*; *Stewart et al., 2000*). Patient-centred care can also be described with two dimensions. The first dimension (i.e., the sharing dimension) refers to the sharing of power, responsibility, and information. The second dimension (i.e., the caring dimension) considers whether the patient's feelings, expectations, and preferences are taken into account in medical decision-making (*Lee et al., 2008a*). Patient-centred care has been associated with positive patient outcomes in many fields by increasing satisfaction, promoting effective communication, reducing medical complaints, decreasing consultation time, improving perceptions on service delivery quality (*Pereira et al., 2013*; *Stone, 2008*; *Hudon et al., 2011*).

With an increase in the realization of patient-centred care's advantages and importance, this approach has attracted attention worldwide and been measured in many countries including the United States, Brazil, and Pakistan (*Haidet et al., 2002*; *Ribeiro, Krupat & Amaral, 2007*; *Ahmad et al., 2015*; *Mudiyanse et al., 2015*). In recent years, medical educators have gradually established clinical practice courses with the purpose of teaching communication skills, professional values, attitudes, and humanistic behaviors to medical

students (*Schmidt, 1998*). In healthcare and medical education, it is important to assess practitioners' attitudes about care (*Pereira et al., 2013*). The Patient-Practitioner Orientation Scale (PPOS) is an instrument that was developed in 1999 in America to assess physicians', medical students', and patients' attitudes regarding whether providers and patients should have equal power and control (*Krupat et al., 1999*; *Street et al., 2003*; *Krupat et al., 2000*). The Patient-Practitioner Orientation Scale (PPOS) contains a four-element patient-centred care model and has been translated into different languages and validated in several countries (*Krupat et al., 1999*; *Kim, 2013*; *Tsimtsiou, Kirana & Hatzichristou, 2014*; *Dockens, Bellon-Harn & Manchaiah, 2016*).

In southwest China, Ting et al. first attempted to use the PPOS to investigate patients' perceptions of patient-centred communication in doctor-patient consultations (*Ting et al., 2016*). Later, Wang et al. improved the Chinese translation and incorporated characteristics of Chinese culture and the Chinese medical and healthcare fields to develop the Chinese-revised Patient-Practitioner Orientation Scale (CR-PPOS), which was based on the original PPOS scale (*Wang et al., 2017*). First, the patient-centred scale was sinicized with the permission of the original author to translate it in a Chinese context. Next, the bilingual PPOS versions were separately sent to five advanced health practitioners in China for further modifications. After that, the Chinese PPOS (C-PPOS) was then retranslated into English and sent back to the original author for confirmation regarding its accuracy. Finally, based on reliability, validity, and discriminative power tests, the C-PPOS was revised to the CR-PPOS. The psychometrics of the CR-PPOS has been validated with a reliability test (test-retest reliability) and validity tests (exploratory factor analysis, confirmatory factor analysis, and internal consistency). In the relevant study, the results were within the acceptable range. Wang et al. conducted a survey with the CR-PPOS in Shanghai, China to investigate preferences towards patient-centred communication among physicians and patients. This is the only known application of CR-PPOS in China (*Wang et al., 2017*). Though it is an innovative tool, the CR-PPOS does not measure medical students' patient-centred attitudes in China. For example, it only explores physicians' and patients' views on patient-centred communication and lacks medical students' views.

Medical students' patient-centred attitudes have been measured in numerous countries. For instance, American scholar Krupat et al. measured first-year US medical students' patient-centred attitudes and found that female students scored higher on patient-centred attitude scales than did male students (*Krupat et al., 1999*). Additional American scholars, Haidet and colleagues, conducted studies on patient-centred attitudes during medical education programs and established that there are gender differences in such attitudes; female medical students have a higher prevalence of these attitudes (*Haidet et al., 2002*). Furthermore, Kheng Hock Lee, in his study on Asian (Indian) students' patient-centred attitudes, also demonstrated the same gender differences (*Lee et al., 2008b*). Therefore, existing studies have reported that there are gender differences in patient-centred attitudes. The present study will advance measurements of Chinese medical students' patient-centred attitudes, and it will explore whether gender and other background factors impact such attitudes (as has been established in countries outside of China). In doing so, the present study will increase an understanding of patient-centred attitudes in China, and it

will provide references for future medical education practices based on China's current doctor-patient relationships and its medical/healthcare policy background.

## METHODS

### Study population

A cross-sectional study was conducted from November 2017 to March 2018 in Heilongjiang Province, Northeast China, which has a population of approximately 38.1 million people. The participants were all fourth-year medical undergraduate students at Harbin Medical University's First, Second, and Fourth Affiliated Hospitals. Harbin Medical University offers two types of the length of schooling, a five-year category and a seven-year category. At this university, the first 3.5 years of study are considered to be 'pre-clinical' (with the occasional clinical practice courses), and the last years (either 1.5 or 3.5 years, depending on the length of schooling) are entirely clinical and provide students with opportunities to interact with patients. During the present study's investigation, the fourth-year students were involved in multiple clinical rotations. The surveyor distributed the questionnaire in person to the students during breaks between two classes. Also, the surveyor informed the students of how to fill in the questionnaire and supervised the entire process until the student completed the questionnaire in order to ensure quality and efficiency. Participation in the survey was voluntary, and the researchers obtained oral informed consent prior to beginning the study and a group-wide oral informed consent was read by the investigators to the participants. Next, all participants completed a self-administered questionnaire within 10–15 min. The data were collected anonymously to ensure participants' confidentiality. Of the 563 distributed questionnaires, 513 were considered valid and the valid response rate was 91.12%.

### Study instrument

The original PPOS includes 18 items (*Krupat et al., 1999*), and each item is presented as a statement in which respondents are asked to indicate their attitudes using a six-point Likert scale, from *strongly agree* to *strongly disagree* (*Krupat et al., 2000*; *Street et al., 2003*). The scale includes the 'Sharing' and 'Caring' dimensions (as described previously). Given the particularities of Chinese culture and the differences in doctor-patient relationships across countries, the present study utilized a survey with the CR-PPOS scale. The CR-PPOS scale is a self-administered questionnaire, and it contains 11 items and two subscales (*Wang et al., 2017*). Of the 11 items, five were used to evaluate caring attitudes and the remaining items were used to evaluate sharing attitudes. The CR-PPOS item scores range from *strongly disagree* = 1 to *strongly agree* = 6. In order to be comparable with foreign PPOS scores, the CR-PPOS score is inverted so that higher scores indicate more patient-centred attitudes. The CR-PPOS average score is from 1 to 6, and scores higher than 3.5 indicate patient-centred attitudes; the rest of the scores represent doctor-centred attitudes.

### Data analysis

Frequencies and percentages were calculated for the demographic characteristics (measured as categorical variables). The demographic information collected in the survey included

gender (male/female), age ($\leq 22/ > 22$), student category (*Yang et al., 2014*) (five-year or seven–year category), without siblings (yes/no), family residence location (city, town, or village), whether the student's parents are doctors (yes/no), and year in which the student first experienced clinical practice (second year and below/ third year and above). We included the variable that measured whether the one-child policy has an impact on the patient-centred attitude of medical students. Likewise, we included students' residential backgrounds (e.g., village or town) because we expect that students from socially closer communities which families are closely connected with each other and have more neighborhood care and concern will be more likely to develop patient-centred attitudes.

The CR-PPOS descriptive statistics for the different genders were analyzed by calculating the means for the total scores. To do so, a T test was used to conduct a difference comparison. A single factor analysis was conducted by comparing the differences in CR-PPOS scores (overall CR-PPOS, 'Caring' subscale, and 'Sharing' subscale), and the scores were examined with T tests for all of the categorical variables listed above (i.e., gender, age, residence, etc.). A one-way analysis of variance (ANOVA) was conducted for the students from clinical hospitals (the First, Second, and Fourth-affiliated hospitals), with statistical significance set at $P < .05$. After one-way analysis, statistically significant variables were screened out and included in a multi-factor regression model for multivariable logistic regression analysis. The social demographic characteristics of gender, clinical hospitals, student category, without siblings, and family residence were taken as covariables. The overall CR-PPOS, "Caring" subscale, and "Sharing" subscale were divided according to median scores (above the median score was the highest level, and the median and below were the lowest). In this study, an overall CR-PPOS score >40 was "patient-centred" and $\leq 40$ was "doctor-centred", a "Caring" subscale score >17 was "patient-centred" and $\leq 17$ was "doctor-centred", and a "Sharing" subscale score >23 was "patient-centred" and $\leq 23$ was "doctor-centred." The dependent variable in the multivariable logistic regression included the following: 1 "patient-centred attitudes" and 0 "otherwise". Odds ratios (ORs) along with 95% confidence intervals were calculated as the measure of association between the outcomes and exposures. All data were inputted manually and organized in Epidata 3.1. For the data analysis, IBM SPSS V.22.0 for Windows was used.

**Ethical approval:** Approval to conduct this study was granted by the Research Ethics Committee of Harbin Medical University.

## RESULTS

### Study sample demographic characteristics

A total of 513 students were surveyed, with 394 (76.8%) students in the five-year clinical category and 119 students (23.2%) in the seven-year clinical category. A majority of the participants were female (60.4%). The participants' ages ranged from 19 to 25, and the mean age was 22. Nearly two-thirds of the participants did not have brothers (sisters). Also, 219 (42.7%) students lived in a city. More than 90% of respondents' parents were not doctors. In addition, 339 (66.1%) participants first had experience with clinical practices in their third year of medical school or later. Table 1 displays the distribution of students.

**Table 1  Demographic characteristics of the study sample ($n = 513$).**

| Characteristic | Male $n$ (%) | Female $n$ (%) | Total $n$ (%) |
|---|---|---|---|
| Student category | | | |
| Five-year clinical category | 156 (76.8) | 238 (76.8) | 394 (76.8) |
| Seven-year clinical category | 47 (23.2) | 72 (23.2) | 119 (23.2) |
| Age | | | |
| ≤22 | 137 (67.5) | 241 (77.7) | 378 (73.7) |
| >22 | 66 (32.5) | 69 (22.3) | 135 (26.3) |
| Without siblings | | | |
| Yes | 134 (66.0) | 190 (61.3) | 324 (63.2) |
| No | 69 (34.0) | 120 (38.7) | 189 (36.8) |
| Family residence | | | |
| City | 82 (40.4) | 137 (44.2) | 219 (42.7) |
| Town | 69 (34.0) | 106 (34.2) | 175 (34.1) |
| Village | 52 (25.6) | 67 (21.6) | 119 (23.2) |
| Doctor parents | | | |
| Yes | 20 (9.9) | 29 (9.4) | 49 (9.6) |
| No | 183 (90.1) | 281 (90.6) | 464 (90.4) |
| Year of first clinical experience | | | |
| Second year and below | 72 (35.5) | 102 (32.9) | 174 (33.9) |
| Third year and above | 131 (64.5) | 208 (67.1) | 339 (66.1) |

## PPOS scores for different major categories and gender

The average total CR-PPOS score for the entire cohort was $3.63 \pm 0.54$, ranging from 1.00 to 5.55. The average 'Caring' score for the entire cohort was $4.53 \pm 0.82$, and the average 'Sharing' score was $2.88 \pm 0.67$. Figure 1 depicts the distributions of overall CR-PPOS scores by gender. The cross-sectional difference in the 'Caring' subscale scores, the 'Sharing' subscale scores, and the total CR-PPOS scores differed between females and males. Female students had significantly higher overall PPOS, 'Caring', and 'Sharing' scores ($3.69 \pm 0.60$, $4.61 \pm 0.74$, $2.92 \pm 0.63$, respectively) than did male students ($3.54 \pm 0.50$, $4.42 \pm 0.92$, $2.81 \pm 0.72$, respectively). Gender was significantly associated with total CR-PPOS score, 'Caring' subscale score, and 'Sharing' subscale score ($P = 0.002$, $P = 0.009$, $P = 0.051$, respectively).

## Single factor analysis for overall PPOS, 'Sharing' subscale, and 'Caring' subscale scores

Three demographic variables were significantly related to higher overall CR-PPOS scores. Female students, students who studied at the First-affiliated hospital, and students with no siblings all demonstrated significantly higher patient-centred scores than students who were male and students with siblings ($P < 0.05$ for each comparison). There was a slight trend toward higher (i.e., more patient-centred) scores among students who lived in villages but this was not statistically significant ($P = 0.071$).

The analysis of the 'Caring' subscale scores revealed additional associations with the demographic variables. Higher 'Caring' scores indicate that doctors take patients' feelings,

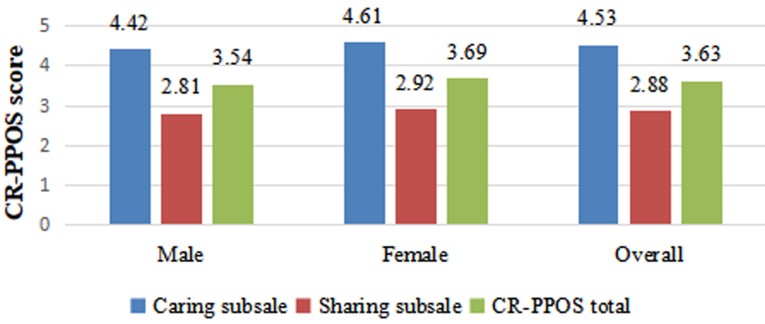

**Figure 1   Distribution of 'Caring' subscale, 'Sharing' subscale and CR-PPOS scores by gender.**

expectations, and preferences into account in their decision-making process more than doctors with lower 'Caring' scores. Female students, students who studied at the First-affiliated hospital, and students in the seven-year clinical category had higher scores than other students ($P < 0.05$). There was a slight trend towards higher 'Caring' subscale scores for students who lived in villages, but this trend was not statistically significant ($P = 0.086$).

Three demographic variables were significantly associated with the 'Sharing' subscale scores. Higher 'Sharing' scores indicate a greater belief that doctors share equal power, control, and information with patients. Students older than 22-years-old, students in the five-year clinical category, and female students had higher scores ($P < 0.05$, $p < 0.05$, and $p = 0.051$, respectively). There was a slight trend toward higher (i.e., more patient-centred) scores for students who studied at the First-affiliated hospital, but this was not statistically significant ($P = 0.061$). The year in which students first experienced clinical practice, as well as the students who have parents that are doctors, had no bearing on patient-centred care attitudes. Table 2 displays these results.

## Multivariable Logistic regression analysis for overall PPOS, 'Sharing' subscale, and 'Caring' subscale scores

Multivariable logistic regressions were used to analyze the factors that influence the students' patient-centred attitude. In the Caring Subscale, those who were at seven-year clinical category and those who studied at the First-affiliated hospital had higher patient-centred attitude (OR = 1.817, 95% CI [1.187∼2.782], OR = 1.817, 95% CI [1.455∼3.009]). There was a slight trend toward female students (OR = 1.415, 95% CI [0.976∼2.053]). In Sharing Subscale, female students, students who were at five-year clinical category and those who studied at the First-affiliated hospital had higher patient-centred attitude(OR = 1.470, 95% CI [1.019∼2.119]; OR = 0.483, 95% CI [0.313∼0.746]; OR = 1.740, 95% CI [1.210∼2.503]). There was a slight trend toward students who didn't have siblings (OR = 1.463, 95% CI [0.984∼2.175]). In total CR-PPOS, students who studied at the First-affiliated hospital had higher patient-centred attitude (OR = 2.286, 95% CI [1.591∼3.284]). There was a slight trend toward female students (OR = 1.438, 95% CI [0.994∼2.079]). Table 3 displays these results.

**Table 2  Total and subscale PPOS scores by student demographic information[*].**

| Characteristic | Caring subscale | Sharing subscale | CR-PPOS total |
|---|---|---|---|
| Gender | | | |
|   Male | $4.42 \pm 0.92$ | $2.81 \pm 0.72$ | $3.54 \pm 0.60$ |
|   Female | $4.61 \pm 0.74$ | $2.92 \pm 0.63$ | $3.69 \pm 0.50$ |
|   P value | **0.009[**]** | **0.051[#]** | **0.002[**]** |
| Student category | | | |
|   Five-year clinical category | $4.47 \pm 0.85$ | $2.93 \pm 0.68$ | $3.63 \pm 0.57$ |
|   Seven-year clinical category | $4.73 \pm 0.69$ | $2.71 \pm 0.61$ | $3.63 \pm 0.46$ |
|   P value | **0.003[**]** | **0.002[**]** | 0.930 |
| Age | | | |
|   $\leq 22$ | $4.49 \pm 0.85$ | $2.85 \pm 0.69$ | $3.59 \pm 0.58$ |
|   $>22$ | $4.61 \pm 0.76$ | $2.93 \pm 0.64$ | $3.69 \pm 0.48$ |
|   P value | 0.523 | **0.004[**]** | 0.138 |
| Clinical hospital | | | |
|   First-affiliated hospital | $4.71 \pm 0.76$ | $2.96 \pm 0.67$ | $3.76 \pm 0.49$ |
|   Second-affiliated hospital | $4.35 \pm 0.89$ | $2.82 \pm 0.69$ | $3.51 \pm 0.59$ |
|   Fourth-affiliated hospital | $4.52 \pm 0.71$ | $2.82 \pm 0.60$ | $3.59 \pm 0.48$ |
|   P value | **0.000[**]** | **0.061[&]** | **0.000[**]** |
| Without siblings | | | |
|   Yes | $4.49 \pm 0.85$ | $2.85 \pm 0.69$ | $3.59 \pm 0.58$ |
|   No | $4.61 \pm 0.76$ | $2.93 \pm 0.64$ | $3.69 \pm 0.48$ |
|   P value | 0.121 | 0.171 | **0.047[**]** |
| Family residence | | | |
|   City + Town | $4.50 \pm 0.83$ | $2.86 \pm 0.67$ | $3.61 \pm 0.56$ |
|   Village | $4.65 \pm 0.77$ | $2.93 \pm 0.66$ | $3.71 \pm 0.49$ |
|   P value | **0.086[&]** | 0.353 | **0.071[&]** |
| Doctor parents | | | |
|   Yes | $4.54 \pm 0.89$ | $2.98 \pm 0.67$ | $3.69 \pm 0.56$ |
|   No | $4.53 \pm 0.81$ | $2.87 \pm 0.67$ | $3.62 \pm 0.54$ |
|   P value | 0.943 | 0.259 | 0.419 |
| Year of first clinical experience | | | |
|   Second year and below | $4.56 \pm 0.80$ | $2.84 \pm 0.65$ | $3.62 \pm 0.53$ |
|   Third year and above | $4.52 \pm 0.83$ | $2.90 \pm 0.68$ | $3.64 \pm 0.55$ |
|   P value | 0.594 | 0.310 | 0.752 |

**Notes.**
[*]All scores are mean scores, $n = 513$.
[**]$P < 0.05$.
[&]$p < 0.1$.
[#]$p = 0.051$.

# DISCUSSION

This is the first study to report on Chinese medical students' attitudes toward patient-centred care. Comparing the Chinese medical students' scores with non-Chinese medical students' scores (measured with the same tool) promotes a more comprehensive understanding of Chinese medical students' patient-centred attitudes. This study's results

**Table 3   Variables influencing total and subscale PPOS scores.**

| Dimension | Variables | OR | 95% CI | *P* Value |
|---|---|---|---|---|
| **Caring Subscale** | | | | |
| | **Gender** | | | |
| | Male | – | – | – |
| | Female | 1.415 | 0.976–2.053 | **0.067**[**] |
| | **Student category** | | | |
| | Five-year clinical category | – | – | – |
| | Seven-year clinical category | 1.817 | 1.187–2.782 | **0.006**[*] |
| | **Clinical hospital** | | | |
| | Second + Fourth affiliated hospital | – | – | – |
| | First affiliated hospital | 2.093 | 1.455–3.009 | **0.000**[*] |
| **Sharing Subscale** | | | | |
| | **Gender** | | | |
| | Male | – | – | – |
| | Female | 1.470 | 1.019–2.119 | **0.039**[*] |
| | **Student category** | | | |
| | Five-year clinical category | – | – | – |
| | Seven-year clinical category | 0.483 | 0.313–0.746 | **0.001**[*] |
| | **Clinical hospital** | | | |
| | Second + Fourth affiliated hospital | – | – | – |
| | First affiliated hospital | 1.740 | 1.210–2.503 | **0.003**[*] |
| | **Without siblings** | | | |
| | Yes | – | – | – |
| | No | 1.463 | 0.984–2.175 | **0.060**[**] |
| **CR-PPOS total** | | | | |
| | **Gender** | | | |
| | Male | – | – | – |
| | Female | 1.438 | 0.994–2.079 | **0.054**[**] |
| | **Clinical hospital** | | | |
| | Second + Fourth affiliated hospital | – | – | – |
| | First affiliated hospital | 2.286 | 1.591–3.284 | **0.000**[*] |

Notes.
[*]$P < 0.05$.
[**]$P < 0.07$.

are consistent with the results obtained in most other countries, indicating that female students tend to be more patient-centred than male students (*Haidet et al., 2002*; *Ribeiro, Krupat & Amaral, 2007*; *Lee et al., 2008b*).

In general, Chinese medical students tended toward patient-centred attitudes, as indicated by a mean CR-PPOS score of 3.63 (scores higher than 3.5 indicate patient-centred attitudes), which is higher than Malian (3.38) and Pakistani (3.40) medical students but lower than American (4.57) and Brazilian (4.66) medical students (*Hurley et al., 2018*; *Ribeiro, Krupat & Amaral, 2007*; *Haidet et al., 2002*; *Ahmad et al., 2015*). The Chinese medical students' 'Sharing' subscales (2.88) were generally lower than Malian (3.04), Pakistani (3.18), and Brazilian (4.10) medical student scores. However, the Chinese
medical students' 'Caring' subscales (4.53) were generally lower than Brazilian students (5.20) but higher than Pakistani (3.63) and Malian (3.68) students. Therefore, the Chinese scores were lower than the scores reported by studies conducted in America and Brazil, but the Chinese scores are nearly equivalent to scores obtained in Pakistan and Mali.

These results might be explained by differences in socio-economic conditions or by religious and cultural differences across countries. In China, the results indicate that, although medical students generally have mid-level patient-centred attitudes, they have higher preferences for 'Caring' than for sharing information with patients and engaging in decision-making with patients. The difference in "Caring" versus "Sharing" subscales might be explained by different cultural values, because in Asian cultures, patients prefer doctors who are more likely to make "family-based" or "doctor-based" decisions in the process of diagnosis and treatment. This differs from the culture in Western countries, where patients prefer that the doctors share veracious 'breaking bad news', such as managing end-of-life care. It is a similar to Africa's cultures that have similarly seen higher caring scores as compared to sharing (*Lee et al., 2008b*; *Searight & Gafford, 2005*; *Tai & Tsai, 2003*; *Hurley et al., 2018*). In China, *Ting et al. (2016)* found that patients were more likely to rely on doctors to control consultations, decision-making, and information distribution; also, Ting et al. established that patients do not think that they have the knowledge or ability to handle medical issues (*Ting et al., 2016*; *Epstein et al., 2005a*; *Epstein et al., 2005b*). The higher 'Caring' subscales might be due to medical students hoping that doctor-patient relationships can be based on mutual understanding and that patients can be treated in a friendly way in accordance with their psychosocial background information (Ting et al., 2016).

This study showed that there is a difference between female and male students in that female students tend to have more patient-centred attitudes than male students (*Ribeiro, Krupat & Amaral, 2007*). The female students surveyed in this study displayed higher mean scores than male students for the overall CR-PPOS scores and for the 'Sharing' and 'Caring' subscale scores. Women are seen as more empathetic and as having better communitarian orientations than men (*Wahlqvist et al., 2010*). *Hojat et al. (2009)*, in a study on the empathy of medical students, demonstrated that there are differences by gender regarding the development of empathy throughout medical school. Moreover, in communicating with patients, women (more so than men) have greater communication abilities that involve more positive conversations, higher use of emotional conversations, and active consultations with patients, which might mean that they are more acceptable to patients (*Roter, Hall & Aoki, 2002*). Furthermore, these characteristics of female doctors might explain why there are fewer lawsuits against female doctors around the world (*Sandhu et al., 2009*). In general, our results corroborate the findings of these existing studies.

Previous PPOS measurement results have indicated that patient-centred attitudes can be influenced by both investigators' personal characteristics and social environmental factors. In the present study, students at the First-affiliated hospital displayed higher CR-PPOS, 'Caring', and 'Sharing' mean scores for patient-centred care. It is plausible that these results can be explained by patients' varying residences and by the specific economic

conditions at the different affiliated hospitals. Compared with other affiliated hospitals, the First-affiliated hospital patients were largely from local areas, and they therefore harmoniously communicated with doctors due to having similar cultural backgrounds. There is also evidence that patients' attitudes and behaviors exhibited during the clinical encounters can influence doctors' patient-centred preferences (*Lee et al., 2008b*), which might help explain why the patient-centred care scores at the First-affiliated hospital were higher than the scores obtained at the other hospitals.

The seven-year clinical category students had higher 'Caring' subscale scores than did the five-year clinical category students; however, interestingly, they scored lower on the 'Sharing' subscale than did the five-year category students. This might be because the seven-year category medical students were participating in clinical doctor-patient communication courses during the investigation; however, the five-year clinical category students had previously taken such courses before the investigation started. The timing of when a student participates in the practical courses has an impact on the 'Sharing' and 'Caring' dimensions. While taking a doctor-patient communication course in clinical practice, the seven-year clinical category students are more likely to have empathy and to be able to use the patients' perspectives to understand patients and to provide care. Also, these students regard patients as a group that needs to be cared for rather than as a group with equal rights to doctors; therefore, these students then have higher 'Caring' subscale scores but lower 'Sharing' subscale scores compared with the five-year clinical category students.

This study also found that students with siblings have higher Sharing scores. One possible explanation for this findings is that the students with siblings are more likely to share, to communicate, and to pay attention to others' feelings in their daily lives. These explanations help shed light on why these students achieved higher Sharing scores.

## CONCLUSION

In general, our findings indicated that Chinese medical students' attitudes are more patient-centred as measured by the CR-PPOS and 'Caring' subscale scores but less patient-centred as measured by the 'Sharing' subscale scores. Gender has a significant impact on medical students' patient-centred attitudes, as has been reported in other countries. Higher patient-centred attitudes were also associated with seven-year clinical category students, students who studied at the First-affiliated hospital, students from villages, and students without siblings.

Our research indicates that, if medical schools want to bridge the gap between male and female patient-centred attitudes and to improve patient-centred attitudes, more attention should be paid to the doctor-patient communication courses in clinical practice. The patient-centred differences by gender, major categories, and other factors should be incorporated into medical education. We suggest that the Chinese medical schools learn from Sweden's mixed-group education method, in which male students actively role play doctor-patient communication scenarios so that they develop patience and become approachable (even though there are still distinct differences between men and women)

(*Wahlqvist et al., 2010*). Moreover, it is important that medical schools think critically about at which point in the categories they should offer the relevant doctor-patient communication courses. In addition, this study also identifies that medical students' personal characteristics and social environmental factors impact their patient-centred care attitudes. Improving medical students' patient-centred skills via relevant medical education can result in establishing more high-quality medical services in China. However, education reform alone cannot fully achieve patient-centred care; instead, society as a whole and the entire healthcare system also need to affirm the value and significant of patient-centred care before it can be fully realized.

### Limitations and suggestions for future research

When interpreting our findings, we should bear in mind the limitations of our research. Through further research in the future, we will more comprehensively explore the patient-centred attitude of medical students.

1. The duration of the cross-sectional survey may have an impact on the patient-centred attitudes of medical students.
2. As there are fewer CR-PPOS items than PPOS items, there may be some deviation in comparing the patient-centred attitudes of medical students.
3. The analysis only included cross-sectional comparisons that focused on clinical medical students in one city. It would be helpful if future research incorporated longitudinal analyses or follow-up studies on other types of medical students (e.g., dental students).
4. Future related research might also include large sample sizes to increase our understanding of this topic.

## ACKNOWLEDGEMENTS

The authors would like to thank all medical students who responded to the questionnaire.

### Funding

This work was supported by the National Natural Science Foundation of China (Grant Number: 71673073, 71333003), Heilongjiang Postdoctoral Scientific Research Development Fund (LBH-Q18071), Medical Education Research Project (2018) supported by Medical Education Branch of the Chinese Medical Association and the Medical Education Professional Committee of China Association of Higher Education (2018A-N03045), Special project supported by China Postdoctoral Foundation (2016T90318). The funders had no role in study design, data collection and analysis, decision to publish, or preparation of the manuscript.

### Grant Disclosures

The following grant information was disclosed by the authors:
National Natural Science Foundation of China: 71673073, 71333003.
Heilongjiang Postdoctoral Scientific Research Development Fund: LBH-Q18071.

Medical Education Branch of the Chinese Medical Association.
Medical Education Professional Committee of China Association of Higher Education: 2018A-N03045.
China Postdoctoral Foundation: 2016T90318.

## Competing Interests

The authors declare there are no competing interests.

## Author Contributions

- Wei Liu, Weijian Song, Libo Liang and Qunhong Wu conceived and designed the experiments, authored or reviewed drafts of the paper, approved the final draft.
- Yanhua Hao performed the experiments, authored or reviewed drafts of the paper, approved the final draft.
- Xiaowen Zhao and Tao Peng performed the experiments, contributed reagents/materials/analysis tools, prepared figures and/or tables, approved the final draft.
- Yuxin Xue, Siyi Tao, Zheng Kang, Ning Ning, Lijun Gao and Yu Cui analyzed the data, prepared figures and/or tables, approved the final draft.

## Human Ethics

The following information was supplied relating to ethical approvals (i.e., approving body and any reference numbers):

Approval to conduct this study was granted by the Research Ethics Committee of Harbin Medical University.

## Data Availability

Raw data are available as a Supplemental File.

## Supplemental Information

Supplemental information for this article can be found online at http://dx.doi.org/10.7717/peerj.7896#supplemental-information.

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
