# Peer review of "Gender differences on medical students’ attitudes toward patient-centred care: a cross-sectional survey conducted in Heilongjiang, China"

_PeerJ, doi:10.7717/peerj.7896_

## Round 0.1 · original submission · Major Revisions

There are a number of technical issues in the manuscript that would need to be addressed in order for it to be suitable for publication. Most of them have been pointed out by reviewer #1.

It seems that the manuscript has some English grammatical errors so it would benefit from a thorough copy-edit.

The reviewers and I recommend particularly that you pay attention to the background review (introduction) of the manuscript. It would help readers to better understand the focus of the study.

[]

·

Basic reporting

- The article has some English grammatical errors, of which I have pointed out. Overall, it would benefit from a thorough copy-edit
- Some more citations/justification are needed for certain points (see specific comments to authors)

Experimental design

Original, primary research and defined research question. More detail is needed to justify comparing their locally-adapted scale to results from other countries (e.g. did they perform psychometric validation to their adaptation)? Also, the authors should provide justification as to why a multivariate regression was not performed following the bi-variate comparisons.

Validity of the findings

There are specific areas that would help clarify the validity of the findings in the methods and discussion sections (see general comments for author).

Additional comments

Thank you for the opportunity to review this manuscript. I commend the authors for investigating the issue of patient-centeredness among medical students in their cultural and geographic context. Research like this is needed to help strengthen overall efforts to improve patient-centered care through medical education.
There are a number of technical issues in this article that would need to be addressed in order for it to be suitable for publication.

Abstract:
- Was the CR-PPOS previously validated? If so, it would be good to state in the abstract
- Some reference as to what the authors mean by “program type” would be helpful in interpreting this variable.
- Unclear implications for medical education: Why only bridge the gap? Why not also aim to raise patient-centered attitudes across the board?
INTRO
Line 77: “Traditional medical practices” can hold many meanings. What are the authors referring to? Biomedicine? Or something else
Line 79- It might be argued that patient-centered care is not the more “modern” approach, but is more evidence in pre-biomedical, indigenous systems (example: )
Line 99- I suggest deleting “and so on” as it not specific
Line 102- What do the authors mean by “attracted attention worldwide”? Can they provide specific examples?
Lines 105-106: Why has it become “increasingly” important? What it that about today that makes assessing attitudes more important than in the past?
Lines 107-109: It would be helpful context here to state the original scale was created in the U.S.
Lines 110: The author don’t need to repeat writing out the full acronym again or the original develop year again
Lines: 114-123: Were the psychometrics of the CR-PPOS validated? If so what were the results (do the factors load similarly into two subscales? Is there good internal validity and reliability?) How was the translation conducted? How extensive were the changes to the questions (would the changes affect comparison with results from other countries?)
Lines 122-123: Why is the fact that it only explores physician and patient’s views a limitation? Hasn’t it also been used with medical students? Another limitation would be unstable factor loadings outside of the original U.S. context (Hurley et al 2018 which you cite, and Archer et. al [2014] “Making use of an existing questionnaire to measure patient-centred attitudes in undergraduate medical students: A case study” http://hmpg.co.za/index.php/ajhpe/article/view/498)
Line 130: Where were these Asian students? (China or elsewhere?)
Lines 130-133: Can you specify the gender differences found here? (Did females score higher?)
METHODS
Line 142: This seems like a long time period for cross-sectional study. Why could the survey not be done at one time point? When did most students take the survey, or was it an even distribution across this time period? Might there a shift in overall attitudes at the beginning or end of this period? If so, please discuss in the study limitations.
Line 155: Was oral consent group or individual? Who read participants the informed consent?
Overall: Please explain your choice to not conduct a multivariate regression to examine the joint effect of all variables. It is possible that the effect of one variable might be explained by another.

RESULTS
Line 157-158: This is a very good response rate. How closely does the sample of 513 represent the overall demographics of the students? (e.g. do the authors notice any differences in those who chose to take the survey?)
Line 165: Could the authors specify (or give examples of) what they mean by “particularities”?
Line 179: Please provide some justification for including sibling variable in your investigation. Is there previous research that indicates this might have an effect? Or something about the Chinese culture?
Line 183: What do you mean by “closer”? Geographically or socially? Again, what is the justification for including this variables?
Line 199: Do you mean “siblings” here (instead of brothers [sisters])?
Line 210: Be careful with the wording here- this is not a “pattern of change”, it is a cross-sectional difference.

DISCUSSION
Lines 248-250: Is this finding of females having higher patient-centeredness consistent across ALL other studies you identified in different countries? If not, please temper the language. Also, please provide proper citations here.
Line 253: Do the adjustments to Chinese version (e.g. deletion of some items) allow for such comparisons? If we are not sure, please state that limitation.
Line 268- Can you explain a little more where ‘family-based’ and ‘doctor-based’ decision means and why this is significant to Chinese culture? It is a similar or different to other cultures that have similarly seen higher caring scores as compared to sharing (e.g. the Malian study you cite
Lines 316-318: Unless there are citations to support these statement, use “may” or temper language according to indicate it is merely the author’s viewpoint.

Reviewer 2 ·

Basic reporting

approved.

Experimental design

approved.

Validity of the findings

the validity of the findings approved.

Additional comments

Review comments
Title of the study:
It seems that the present study focused on the effect of gender on the attitude of Chinese medical students about the patient – centered care, therefore the title of the study must reflect the actual aim and purpose of the study and the title must change according to the study aim.
Introduction
It was written in a good sequence and structure and the writing English is good.
Method
Sample
Based on which formula or method the study sample has been calculated. The method for sample selection should explain in detail.
The study Instrument
Detail explanation about the scoring of the instrument, maximum and minimum of the scores, range of the score and interpretation must add to the method.
Data analysis
It’s better that a brief explanation provides about the distribution of the data ( normal or not).
Conclusion
It’s better to add some suggestions for future research.

---

## Round 0.2 · accepted · Accept

The authors did a good job addressing the reviewer's comments

·

Basic reporting

No additional comments

Experimental design

No additional comments

Validity of the findings

No additional comments

Additional comments

The authors did a wonderful job addressing my comments. Congratulations on a high-quality paper.

Reviewer 2 ·

Basic reporting

no comment

Experimental design

no comment

Validity of the findings

no comment

Additional comments

no comment